# Bacterial second messenger 3′,5′-cyclic diguanylate attracts *Caenorhabditis elegans* and suppresses its immunity

Joseph Angeloni[1], Yuqing Dong[1,2], Zeneng Wang[3] & Min Cao [1,2✉]

Cyclic di-nucleotides are important secondary signaling molecules in bacteria that regulate a wide range of processes. In this study, we found that *Caenorhabditis elegans* can detect and are attracted to multiple signal molecules produced by *Vibrio cholerae*, specifically the 3′,5′-cyclic diguanylate (c-di-GMP), even though this bacterium kills the host at a high rate. C-di-GMP is sensed through *C. elegans* olfactory AWC neurons, which then evokes a series of signal transduction pathways that lead to reduced activity of two key stress response transcription factors, SKN-1 and HSF-1, and weakened innate immunity. Taken together, our study elucidates the role of c-di-GMP in interkingdom communication. For *C. elegans*, bacterial c-di-GMP may serve as a cue that they can use to detect food. On the other hand, preexposure to low concentrations of c-di-GMP may impair their immune response, which could facilitate bacterial invasion and survival.

[1] Department of Biological Sciences, Clemson University, 132 Long Hall, Clemson, SC 29634, USA. [2] Institute for Engaged Aging, Clemson University, 2037 Barre Hall, Clemson, SC 29634, USA. [3] Department of Cardiovascular and Metabolic Sciences, Lerner Research Institute, Cleveland Clinic, 9500 Euclid Ave, Cleveland, OH 44195, USA. ✉email: mcao@clemson.edu

Bacteria are able to communicate with one another through a process known as quorum sensing (QS). This process mainly relies on the production and secretion of specific autoinducers (AIs), and allows the cells to make group decisions based on cell-density[1]. While the AIs are extracellular signaling molecules, bacteria also use a range of nucleotide-based intracellular signaling molecules known as second messengers to regulate physiological responses to cope with a changing environment. Molecules such as cyclic adenosine 3′,5′-monophosphate (cAMP) and guanosine pentaphosphate or tetraphosphate ((p) ppGpp) have been well studied for almost 50 years[2,3]. In the past two decades, the field of cyclic dinucleotides (CDNs) is expanding and has attracted more attention in different areas of research. 3′,5′-cyclic diguanylate (c-di-GMP) was the first identified CDN and has been extensively studied since its discovery. Initially characterized as an activator of cellulose synthase in *Acetobacter xylinum*[4], c-di-GMP is now known as a ubiquitous bacterial signal that regulates a variety of physiological processes in Gram-negative bacteria such as motility, biofilm formation, virulence, and transmission between hosts[5]. Its role in modulating flagellar motility, exopolysaccharide production, hyphae formation, etc., has also been reported in a few Gram-positive bacteria[6]. The second CDN, 3′,5′-cyclic diadenylate (c-di-AMP), is mainly found in Gram-positive bacteria. It has been characterized as an essential signaling molecule that regulates cell wall homeostasis, potassium ion channels, DNA integrity, as well as biofilm formation and virulence[7]. Adenosine-guanosine-3′,3′-cyclic monophosphate (cGAMP) is the newest addition to the CDN list that

has only been identified in very few bacteria. In *Vibrio cholerae* (*V. cholerae*), cGAMP plays a role in efficient intestinal colonization[8], and in *Geobacter*, it controls exoelectrogenesis[9].

As more research has been conducted, it was discovered that the role of these signaling molecules is not limited to communication among bacterial cells. Bacteria and their eukaryotic hosts can also communicate with each other via these signaling molecules. This so-called interkingdom communication has recently become an expanding field of research with broad implications. Bacteria can sense and respond to mammalian hormones. For example, the classic stress hormones adrenaline and noradrenaline can induce bacterial growth and virulence expression. Conversely, bacterial AIs (specifically the acyl-homoserine lactones), can enter the mammalian cells and modulate host immune response and promote apoptosis[10]. Bacterial CDNs also have immunomodulation functions. It was reported that *Listeria monocytogenes* secrets c-di-AMP through a multidrug efflux pump and activates host type I interferon[11]. Whether c-di-GMP or cGAMP is secreted by bacteria is not known at this time. However, these signals can be detected by the endoplasmic-reticulum-resident protein STING (stimulator of interferon genes) in humans and mediate the type I interferon immune response[12,13].

In the aforementioned cases, bacterial signals need to enter the host cells in order to elicit a specific response. Recent studies of bacteria-*C. elegans* interactions illustrated that bacterial AIs could function as chemical cues or odors that affect worms' behavior. Beale et al. first reported that *C. elegans* senses *Pseudomonas aeruginosa* acyl-homoserine lactones and are attracted by the signal. They also showed that *C. elegans* could acquire relatively long-lasting memory to avoid the AIs if they were pre-exposed to them for a short period[14]. *V. cholerae*, the etiologic agent of cholera, possesses multiple QS signaling molecules. CAI-1 and AI-2 are the two well-studied autoinducers. The CAI-1 molecule was found to chemoattract *C. elegans*, and it is sensed through the amphid AWC[ON] neuron[15]. In this study, we show that in addition to CAI-1, c-di-GMP is another chemoattractant that can be sensed by *C. elegans* by both AWC[ON] and AWC[OFF] neurons. Sensing c-di-GMP elicits a series of signal transduction pathways in the host, which leads to a reduced innate immune response and a shortened lifespan by affecting the activity of two key stress response transcription factors, SKN-1 and HSF-1.

## Results

### *C. elegans* are attracted to *V. cholerae* by signals other than the QS autoinducers.

In a previous study, we reported that water-soluble cranberry extract protects *C. elegans* from killing by different pathogenic bacteria, including *V. cholerae*, *Pseudomonas aeruginosa*, *Salmonella typhimurium*, *Enterococcus faecalis*, and *Staphylococcus aureus*[16]. During these killing assays, an interesting phenomenon caught our attention. Worms fed on *V. cholerae* seemed to be allured by this type of bacterium and remained in the bacterial lawn until death. In contrast, worms fed on the other four pathogens showed a scattered behavioral pattern on the test plates. It is known that *C. elegans* can sense and avoid bacterial pathogens, but in our study, the worms were lured to a pathogen that significantly decreases their lifespan. A simple choice index (CI) assay was performed with the N2 wild type *C. elegans* over the course of 2 h to test whether the worms prefer the pathogenic *V. cholerae* wild type (wt) strain C6706 rather than *E. coli* OP50, the common food source for *C. elegans* used in the lab. As shown in Fig. 1a, the worms were readily attracted to *V. cholerae* (wt) compared to *E. coli* OP50.

Previously, Werner KM et al. reported that *V. cholerae* QS CAI-1 is a chemoattractant sensed by *C. elegans* amphid sensory

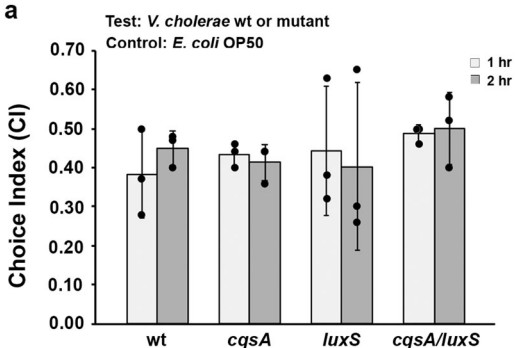

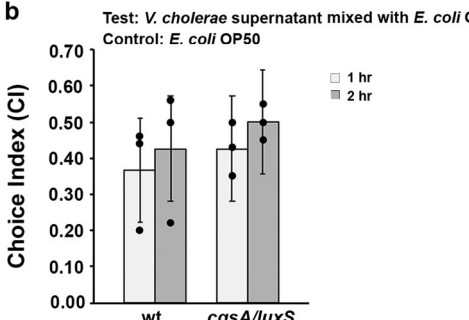

**Fig. 1 C. elegans prefer V. cholerae to E. coli OP50. a** Preference towards *V. cholerae* wild type (wt) and the *luxS−*, *cqsA−*, *cqsA−/luxS−* mutant strains over *E. coli* OP50 over the course of 2 h. **b** Preference towards *E. coli* OP50 mixed with the supernatant of *V. cholerae* wt or the *cqsA−/luxS−* mutant over *E. coli* OP50 over the course of 2 h. All choice indices were calculated from three independent assays, and error bars are standard error of the mean representing 95% confidence intervals. Individual values are shown as black dots. *p* values were calculated by using one-way balanced ANOVA in **a** and using the unpaired Student *T*-test in **b**. In all cases, the *p* values are larger than 0.05, indicating no significant difference among the groups.

neuron AWC[ON15]. To test whether other signal molecules are involved in chemoattraction, we utilized three mutant strains: $cqsA^-$ (no CAI-1 production), $luxS^-$ (no AI-2 production), and a $cqsA^-/luxS^-$ double deletion. Each mutant was tested against *E. coli* OP50 using the standard CI assay. *C. elegans* preference was observed towards all three mutant strains (Fig. 1a), conveying that signals other than the two autoinducers play a role. Interestingly, the choice index of any of the mutant strains over *E. coli* was comparable to that of the wild type over *E. coli*. This observation implies that rather than CAI-1, other signal molecules play a predominant role in attracting *C. elegans*. Next, we mixed the supernatant of the overnight culture of wild type *V. cholerae* and the $cqsA^-/luxS^-$ double mutant with *E. coli* OP50, respectively, and tested worms' preference over *E. coli* OP50. As expected, worms were attracted to *E. coli* OP50 mixed with the supernatant from either the wild type or the double mutant strain (Fig. 1b). This result confirms the existence of other chemoattraction signals present in the *V. cholerae* cell-free supernatant.

**C-di-GMP is the major signaling molecule that attracts *C. elegans*.** QS autoinducers and the cyclic dinucleotides are essential signaling molecules that regulate numerous physiological functions in bacteria. Unlike the QS autoinducers, which are secreted into the environment, the cyclic dinucleotides are generally known as intracellular second messengers. We questioned whether c-di-GMP and cGAMP, the two cyclic dinucleotides characterized in *V. cholerae*, could function as extracellular signals that attract *C. elegans*. To test this, we purchased pure solutions of c-di-GMP and cGAMP. CI assays were conducted as usual, except that *E. coli* OP50 was mixed with two concentrations (0.1 and 1 nM) of each cyclic dinucleotide to observe preference. These concentrations were tested because it was theorized that low levels of these molecules would be present outside of the cells. Figure 2a shows that at 1 nM, both c-di-GMP and cGAMP were able to trigger an attractive behavior in *C. elegans*. As a control, c-di-AMP (1 nM) was also tested in the same assay. In contrast to c-di-GMP and cGAMP, c-di-AMP caused an apparent avoidance in *C. elegans*. We also tested similar signaling molecules, GMP and cGMP, and showed that neither cGMP nor GMP could cause any chemoattraction or repulsion at concentrations similar to what elicited a response from c-di-GMP and cGAMP (Fig. 2a).

We next harvested the supernatant and the cell pellet from the wild type *V. cholerae* overnight culture (~ $10^9$ CFU/ml) by centrifugation, and measured the concentration of c-di-GMP and cGAMP from the two portions using liquid chromatography-mass spectrophotometry (LC-MS) assay. Pure c-di-GMP and cGAMP solutions were used as standards for this assay. As shown in Fig. 2b, c-di-GMP was detected in both the cell lysate and the supernatant. The concentration was calculated as $177.3 \pm 49.4$ nM in the cell lysate and $19.3 \pm 4.6$ nM in the supernatant. No cGAMP was detected in the cell lysate or the supernatant. These results suggest that c-di-GMP may be the additional signal that attracts *C. elegans* towards *V. cholerae*.

To further confirm the role of c-di-GMP in chemoattraction, we sought to manipulate the concentration of c-di-GMP in *V. cholerae*. C-di-GMP is synthesized by the diguanylate cyclases containing a GGDEF domain and degraded by the phosphodiesterases containing either an EAL or HD-GYP domain[17]. pAT1568, a plasmid that overexpresses the phosphodiesterase (EAL domain) of the *vieA* gene from the L-arabinose-inducible pBAD promoter[18], was introduced into the $cqsA^-/luxS^-$ strain by electroporation. The resulting transformant was grown overnight in the presence of 0.2% L-arabinose to induce phosphodiesterase expression and was used in CI assays against *E. coli* OP50. As a

control, the empty vector pBAD33 was also introduced into the $cqsA^-/luxS^-$ strain and cultivated in a similar fashion. Figure 2c shows the CI result. With pAT1568 (i.e., decreased c-di-GMP level), *C. elegans* was no longer attracted to the $cqsA^-/luxS^-$ strain (choice index ≈ 0), while with the empty vector pBAD33, a strong preference was still observed. This observation revealed the critical role of c-di-GMP production and presence in the chemoattractive behavior of *C. elegans*. Combining the results from Fig. 2a, b, we consider that c-di-GMP is the major signaling molecule that attracts *C. elegans*.

**C. elegans senses c-di-GMP through the AWC neurons and the cGMP-gated TAX-2/TAX-4 channel.** The chemosensory system in *C. elegans* allows the organism to detect food, develop, avoid danger, mate, etc. There are 11 pairs of amphid chemosensory neurons in *C. elegans*. Of these, the ASE gustatory neurons are known to sense salts and water-soluble attractants, and the AWA and AWC olfactory neurons are required to sense attractants with volatile odors. Different chemical signals are sensed by different G protein-coupled chemoreceptors in the chemosensory neurons, and then passed to two major signal transduction sensory channels, the cGMP-gated TAX-2/TAX-4 channel and the lipid-sensing OSM-9/OCR-2 TRPV channel[19]. It was reported that *C. elegans* senses *V. cholerae* CAI-1 through the AWC[ON] neuron and uses the TAX-2/TAX-4 channel[15]. We hypothesized that the same neuronal sensory pathway could be involved in sensing c-di-GMP. To test this, different *C. elegans* mutant strains were assessed to determine if they were able to similarly respond to c-di-GMP as the wild type strain. The *tax-2/tax-4* mutant was no longer attracted to c-di-GMP, while the *osm-9/ocr-2* mutant was still attracted, signifying that the cGMP-gated TAX-2/TAX-4 channel is required (Fig. 3). Mutation to *ceh-36* causes defects in developing functional AWC and ASEL neurons, and mutation to *che-1* causes an inability to develop functional ASEL and ASER neurons. Mutations to *nsy-5* and *nsy-1* cause an inability to develop the AWC[ON] and AWC[OFF] neurons, respectively. Figure 3 shows that loss of functional AWC or ASEL neuron (*ceh-36*) kept the worms from the attraction, and worms without the functional ASEL and ASER neuron (*che-1*) were still attracted, indicating the AWC neuron is involved. Further assay with the *nsy-5* and *nsy-1* mutants revealed that the AWC[ON] and AWC[OFF] neurons are both necessary because no preference was observed when studying these mutants. Based on these results, it can be concluded that *C. elegans* senses c-di-GMP through the AWC[ON] and AWC[OFF] neurons and the cGMP-gated TAX-2/TAX-4 channel.

**C-di-GMP suppresses the innate immunity of *C. elegans*.** Since it has been shown that c-di-GMP can cause initial attraction of *C. elegans* over the course of a couple of hours, it was next of interest to investigate if detection of this signal molecule plays a role in the health of the host. Lifespan assays were performed in which NGM-FUDR media was supplemented with 1 nM of c-di-GMP to test the lifespan of *C. elegans* N2, when exposed to c-di-GMP throughout their entire life. The results in Fig. 4a and Table 1 show that the presence of c-di-GMP significantly affects the lifespan of *C. elegans* N2 in that worm populations typically are living much shorter compared to control. The mean number of days lived of N2 worms at 25 °C when exposed to c-di-GMP was $11.30 \pm 0.48$, whereas nonexposure was $13.66 \pm 0.39$, an 17.0% decrease ($p < 0.001$). The lifespan shortening effect is specific to the c-di-GMP molecule, as neither cGAMP nor c-di-AMP at 1 nM affected the normal lifespan of N2 worms (Fig. 4a and Table 1). We also found that the presence of live bacteria is requisite, as c-di-GMP only caused a slight decrease in the

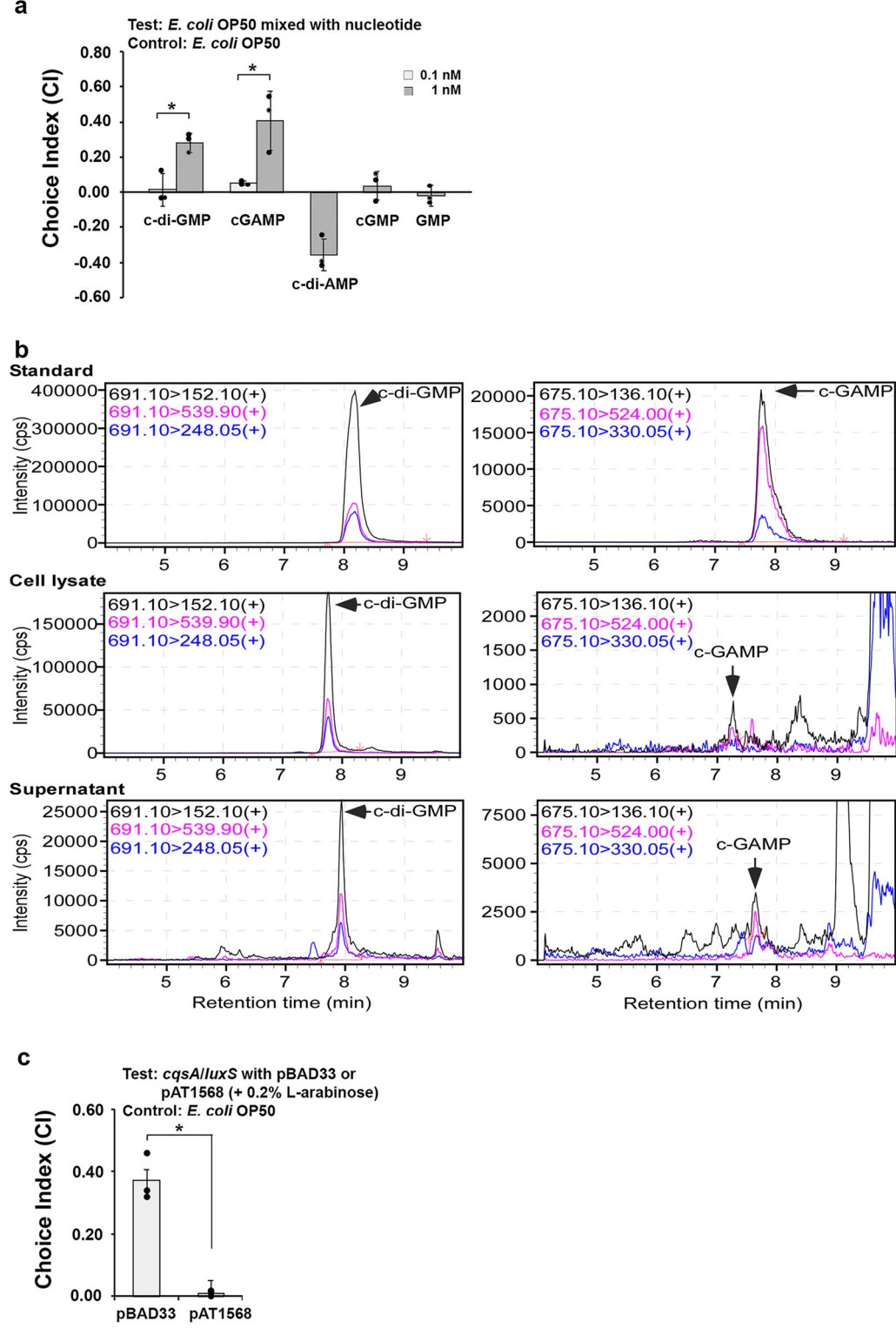

**Fig. 2 C-di-GMP is the major signal in *V. cholerae* supernatant that chemoattracts *C. elegans*. a** Preference towards *E. coli* OP50 supplemented with 1 nM c-di-GMP and cGAMP. Avoidance behavior was observed with 1 nM c-di-AMP. No preference was observed with either cGMP or GMP. Results are the average of three independent experiments, and error bars are standard error of the mean representing 95% confidence intervals. Individual values are shown as black dots. Unpaired Student *T*-test was used to calculate the *p* values (0.1 nM vs. 1 nM). *p < 0.05. **b** LC-MS assay of c-di-GMP and cGAMP from the cell lysate and supernatant of wild type *V. cholerae* overnight culture. c-di-GMP and cGAMP were monitored by electrospray ionization mass spectrometry in positive mode with multiple reaction monitoring (MRM) at the transitions of *m/z* 691.10 → 152.10, 691.10 → 539.90, 691.10 → 248.05, and *m/z* 675.10 → 136.10, 675.10 → 524.00, 675.10 → 330.05, respectively. **c** Decreasing intracellular c-di-GMP (*cqsA⁻/luxS⁻* with pAT1568) abolishes *C. elegans* preference towards *V. cholerae*. Results are the average of three independent experiments, and error bars are standard error of the mean representing 95% confidence intervals. Individual values are shown as black dots. *p* values were calculated by using the unpaired Student *T*-test (pAT1568 vs. pBAD33). *p < 0.05.

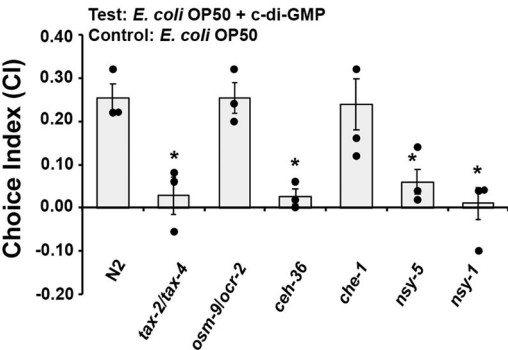

**Fig. 3 *C. elegans* senses c-di-GMP through the AWC neurons and uses the Tax-2/Tax-4 channel.** Choice index (CI) results are the average of three independent experiments, and error bars are standard error of the mean representing 95% confidence intervals. Individual values are shown as black dots. *p* values were calculated by using the unpaired Student *T*-test (each mutant vs. N2). *$p < 0.05$.

lifespan when the N2 worms were fed on heat-killed *E. coli* OP50 (Fig. 4b and Table 1).

Younger worms have robust grinders that can break down bacteria effortlessly, thus live bacteria are often absent from the lumen. There is a decline in grinder function as the worms age, which allows bacteria to easily escape to the lumen of the gut and proliferate[20,21]. Effective curb of intestinal bacterial accumulation is an indicator of strong gut immunity, and also an important causative factor of lifespan determination[20,21]. We therefore compared the gut colonization of *E. coli* OP50 from day-6 (equivalent to middle age) N2 worms in the presence and absence of c-di-GMP. A substantial increase of bacterial number was observed with the addition of 1 nM c-di-GMP (Fig. 4c). Incubating *E. coli* OP50 with the same concentration of c-di-GMP in vitro neither increases the growth rate nor enhances bacterial surface attachment (Supplementary Figs. 1 and 2). Moreover, when *E. coli* OP50 was cultured with 1 nM c-di-GMP overnight and then fed to N2 worms (no c-di-GMP added to the assay plates), the resulting lifespan remains the same as the control (Supplementary Fig. 3). These observations indicate that the increased gut colonization is largely attributed to a weakened gut immunity in the host.

We speculated that the expression of *C. elegans* innate immune response genes might be inhibited by c-di-GMP based on previous results. To examine this, we selected a few innate immune genes (*C23G10.1, clec-46, clec-71, col-41, dct-5, fmo-2, pqn-5,* and *dod-22*) that are reported to be upregulated during bacterial infections[22,23] and analyzed their expression by qRT-PCR. Figure 5 shows that when synchronized L4-stage N2 worms were exposed to 1 nM c-di-GMP for only 10 min, expression of these genes was generally reduced 2-fold to 5-fold except for *dod-22*, which decreased by 1.6-fold. This was consistent with our hypothesis that the presence of this signaling molecule is modulating innate immunity.

**C-di-GMP acts through SKN-1 and HSF-1 to impact immunity and lifespan.** The FOXO family protein DAF-16, the SKN-1 protein, and the heat shock protein HSF-1 are the three major stress response transcription regulators in *C. elegans*. DAF-16, the downstream transcription factor of the insulin/insulin-like growth factor-1 signaling (IIS) pathway, plays a key role in modulating longevity and immunity[24–26]. SKN-1 is the downstream effector of the major immune-signaling p38 mitogen-activated protein kinase (MAPK) pathway and controls numerous genes involved in stress response and lifespan regulation[27].

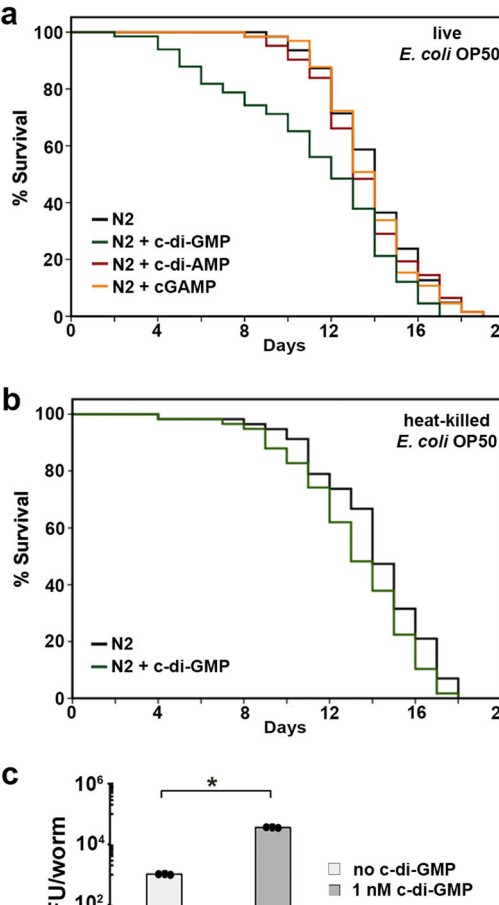

**Fig. 4 C-di-GMP affects *C. elegans* lifespan and gut colonization.** **a** Lifespan of N2 worms fed with live *E. coli* OP50 in the presence of c-di-GMP, c-di-AMP, and cGAMP. **b** Lifespan of N2 worms fed with heat-killed *E. coli* OP50 in the presence of c-di-GMP. The lifespan assays were repeated in at least three independent trials with similar results. The data shown in the figure are representatives from one of the trials. Quantitative data and statistical analyses for these trials are included in Table 1. **c** Average CFU/N2 worm (aged to Day 6) when fed with *E. coli* OP50 at 25 °C in the presence or absence of 1 nM c-di-GMP. Results are the average of three independent experiments, and error bars are standard error of the mean representing 95% confidence intervals. Individual values are shown as black dots. *p* values were calculated by using the unpaired Student *T*-test (N2 vs. N2 + c-di-GMP). *$p < 0.05$.

HSF-1 is another versatile transcription factor that regulates a multitude of genes involved not only in stress response but also in development, metabolism, as well as lifespan and immunity modulation[28–30]. To test whether c-di-GMP acts through these regulators, lifespan assays in the presence and absence of c-di-GMP were conducted in mutant *daf-16* and *hsf-1* strains. In the case of *skn-1*, RNA interference (RNAi) was utilized to knock down the gene expression, and the empty vector pL4440 was used in parallel to serve as the control. We reasoned that if c-di-GMP acts through a particular transcriptional regulator to shorten the lifespan, mutating or knocking down expression of the gene will eliminate the effect. Figure 6a–c and Table 1 show the results. Supplementation of 1 nM of c-di-GMP still reduced the lifespan of the *daf-16* mutant strain, and the percentage of decrease is comparable to that in the wild type N2. Lifespan was not affected

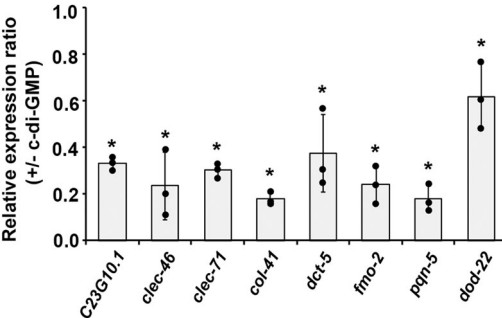

**Fig. 5 qRT-PCR analysis of *C. elegans* innate immunity genes in response to 1 nM c-di-GMP.** Results are the average of three independent experiments, and error bars are standard error of the mean representing 95% confidence intervals. Individual values are shown as black dots. *p* values were calculated by using the unpaired Student *T*-test (each gene vs. *act-1*). *$p < 0.05$.

by c-di-GMP in the *hsf-1* mutant and the *skn-1* RNAi knockdown strains, meaning that SKN-1 and HSF-1 are required for c-di-GMP to exert its effect.

Next, two representative target genes of DAF-16 (C10G11.5 and F52H3.5), two of SKN-1 (*gcs-1* and *gst-4*), and two of HSF-1 (*hsp-16.2* and *hsp-70*) were measured for their expression levels in the presence and absence of c-di-GMP. Figure 6d shows that c-di-GMP triggered downregulation of the SKN-1 and HSF-1 target genes but not the DAF-16 targets. These results suggest that sensing c-di-GMP by the chemoreceptors in the sensory neuron elicited signal transduction, that eventually led to reduced activity of SKN-1 and HSF-1.

**p38 MAPK(PMK-1) and RTK-Ras-ERK MAPK (MPK-1) are required in the c-di-GMP-elicited signal transduction pathway.** MAP kinases are central components of a series of signal transduction pathways that control many vital cellular processes. MAP kinases in mammals are grouped into three families: p38/SAPKs (stress-activated protein kinases), JNKs (Jun amino-terminal kinases), and ERKs (extracellular-signal-regulated kinases)[31]. The three corresponding MAPKs in *C. elegans* are PMK-1, JNK-1, and MPK-1. Activity and cellular localization of SKN-1 and HSF-1 are modified by phosphorylation. Phosphorylated SKN-1 and HSF-1 are translocated into the nucleolus where they activate their target genes. Previous studies reported that either PMK-1[32] or MPK-1[33] possess the ability to directly phosphorylate SKN-1 at the same site. The kinases that phosphorylate HSF-1 remains elusive. Using the *pmk-1* and *jnk-1* mutant strains as well as the *mpk-1* RNAi, we examined whether these MAPKs are involved in the signal transduction pathway elicited by c-di-GMP. Lifespans of these mutant worms were compared in the presence and absence of 1 nM c-di-GMP and shown in Fig. 7a–c and Table 1. In the *pmk-1* mutant and the *mpk-1* RNAi worms, adding c-di-GMP in the medium no longer reduced the lifespan as compared to that in the N2 or N2/pL4440 worms. While in the *jnk-1* mutant, a 10% decrease in lifespan was observed. Our data suggest that the two MAP kinases, PMK-1 from the p38 MAPK pathway and MPK-1 from the RTK-Ras-ERK pathway, are required in the c-di-GMP-elicited signal transduction.

## Discussion
Using the *V. cholerae-C. elegans* interaction model, we discovered that c-di-GMP acts as an interkingdom communication signaling molecule to attract *C. elegans* and suppress the host's innate immunity. Sensing c-di-GMP is through the olfactory AWC$^{ON}$ and AWC$^{OFF}$ neurons and the cGMP-gated TAX-2/TAX-4

channel. This is different from the case of sensing *V. cholerae* CAI-1, which only requires the AWC$^{ON}$ neuron[15]. Unlike the gustatory neurons, olfactory neurons are able to detect long-range signals, usually at nanomolar concentrations. In consensus with this, the concentration of c-di-GMP from the overnight *V. cholerae* culture was measured to be around 20 nM.

Though the two species may not intersect in nature, we think our findings could have universal implications. Being a soil-dwelling nematode, *C. elegans* feeds on bacteria and has coevolved with bacteria over the years, and thus has developed a particularly intimate relationship with bacteria. We believe that the production of c-di-GMP by bacteria, or its presence outside of the bacterial cell, acts as a cue attracting *C. elegans* towards a food source. That initial response and attraction from the worms by c-di-GMP could signal a favorable environment. While this attraction may decrease immune response initially, degradation and weakening of this signal over time could lead to a return to baseline immunity under noninfectious conditions in vivo. However, when this signal is produced by a pathogen, such as *V. cholerae*, this initial attraction could be all it takes for *C. elegans* to begin feeding and thus become infected. Pathogens such as *V. cholerae* and *P. aeruginosa* possess a powerful signaling network that utilizes c-di-GMP for its success. C-di-GMP has been considered to be prevalent in Gram-negative bacteria, but its existence and function in Gram-positive bacteria have been gradually revealed in recent years, especially in those commonly found in the soil, such as *Bacillus* spp.[34–37], *Streptomyces coelicolor*[38], *Clostridium difficile*[39], and *Mycobacterium smegmatis*[40]. With the attraction of *C. elegans* towards low concentrations of c-di-GMP, this can allow it to find numerous bacteria in the environment, Gram-positive or Gram-negative, pathogen or not.

Our observation seems contradictory to the current knowledge that CDNs trigger the innate immune system by activating interferon genes[12]. However, we do not think it is a contradiction. *C. elegans* lacks the c-di-GMP receptor STING[41] and the robust interferon system, therefore c-di-GMP will not trigger the canonic STING pathway in the *C. elegans* model. Only within the last decade has significant research been conducted investigating the binding of bacterial CDNs to the human STING receptor. Initially, this was implicated in the innate immune response to pathogens, but more recently, it has been shown to detect tumor-derived DNA and direct antitumor immunity through the versatile type I IFN[42]. The human STING receptor has a higher binding affinity with cGAMP and c-di-GMP than it does to c-di-AMP[43]. Its affinity is even higher when binding endogenous mammalian cGAMP, which possesses a 2′-5′ phosphodiester linkage rather than the microbial 3′-5′ linkage[13]. Through these studies, it was found that STING has the affinity to bind to the bacterial c-di-GMP at a concentration of about 4.4 μM, which initiates the downstream signal transduction. Efforts have been made in attempts to modify STING for higher affinity, and while this proved successful, the concentration of c-di-GMP needed for binding was still high, $0.461 \pm 0.053$ μM[43].

In our study, we found that c-di-GMP, as low as 1 nM, is being sensed by *C. elegans*, and this decreases its immune response. We also found that this behavioral response occurred at similar concentrations towards microbial cGAMP. Although not uncovered in this work, there is a possibility that *C. elegans* may possess a receptor with a higher affinity for CDNs than the mammalian STING. If this were the case, finding this receptor and studying its structure will provide valuable information to the STING research field. The ability to bind CDNs at nanomolar concentrations would remarkably improve STING's sensitivity to foreign and cancer cell-derived DNA. This is an exciting new area of research, and it has the possibility to be far-reaching, especially in the fight against cancer and improving immunotherapies.

**Table 1 The lifespan (days) of N2 *C. elegans* and various mutant strains at 25 °C in the absence and presence of 1 nM of c-di-GMP.**

| Strain | Mean ± SE (Day) | Median (Day) | # of worms | *p*-value | % change |
|---|---|---|---|---|---|
| N2[a] | 13.66 ± 0.39 | 14.00 | 64 | | |
| N2 + c-di-GMP[a] | 11.30 ± 0.48 | 12.00 | 66 | <0.001 | −17.0 |
| N2 + c-di-AMP[a] | 13.53 ± 0.30 | 13.00 | 62 | 0.583 | −1.2 |
| N2 + cGAMP[a] | 13.71 ± 0.26 | 14.00 | 65 | 0.577 | 0.1 |
| N2 (heat-killed OP50)[b] | 14.02 ± 0.37 | 14.00 | 57 | | |
| N2 + c-di-GMP (heat-killed OP50)[b] | 13.14 ± 0.38 | 13.00 | 58 | 0.063 | −6.3 |
| N2[c] | 13.70 ± 0.45 | 14.00 | 59 | | |
| N2 + c-di-GMP[c] | 11.20 ± 0.48 | 11.00 | 60 | <0.001 | −18.2 |
| *daf-16(mgDf50)*[c] | 8.46 ± 0.22 | 8.00 | 61 | | |
| *daf-16(mgDf50)* + c-di-GMP[c] | 7.20 ± 0.23 | 7.00 | 61 | <0.001 | −14.9 |
| EV[d] | 12.94 ± 0.36 | 13.00 | 62 | | |
| EV + c-di-GMP[d] | 10.93 ± 0.36 | 11.00 | 60 | <0.001 | −15.5 |
| *skn-1* RNAi[d] | 11.20 ± 0.29 | 11.00 | 61 | | |
| *skn-1* RNAi + c-di-GMP[d] | 11.75 ± 0.32 | 12.00 | 60 | 0.080 | 4.9 |
| N2[e] | 12.84 ± 0.41 | 13.00 | 62 | | |
| N2 + c-di-GMP[e] | 10.13 ± 0.41 | 11.00 | 62 | <0.001 | −21.1 |
| *hsf-1(sy441)*[e] | 6.31 ± 0.20 | 7.00 | 61 | | |
| *hsf-1(sy441)* + c-di-GMP[e] | 6.10 ± 0.21 | 6.00 | 61 | 0.495 | −3.3 |
| N2[f] | 13.40 ± 0.41 | 14.00 | 60 | | |
| N2 + c-di-GMP[f] | 10.37 ± 0.44 | 11.00 | 60 | <0.001 | −22.6 |
| *pmk-1(km25)*[f] | 12.08 ± 0.52 | 13.00 | 60 | | |
| *pmk-1(km25)* + c-di-GMP[f] | 11.95 ± 0.54 | 13.00 | 61 | 0.950 | −1.1 |
| N2[g] | 14.03 ± 0.37 | 14.00 | 59 | | |
| N2 + c-di-GMP[g] | 11.53 ± 0.41 | 12.00 | 59 | <0.001 | −17.8 |
| *jnk-1(gk7)*[g] | 12.43 ± 0.29 | 12.00 | 60 | | |
| *jnk-1(gk7)* + c-di-GMP[g] | 11.17 ± 0.28 | 11.00 | 59 | 0.003 | −10.1 |
| EV[h] | 12.88 ± 0.40 | 13.00 | 60 | | |
| EV + c-di-GMP[h] | 11.15 ± 0.36 | 11.00 | 60 | <0.001 | −13.4 |
| *mpk-1* RNAi[h] | 11.40 ± 0.30 | 11.00 | 60 | | |
| *mpk-1* RNAi + c-di-GMP[h] | 11.63 ± 0.31 | 12.00 | 59 | 0.518 | 2.0 |

All the worms were fed on live *E. coli* OP50 unless otherwise noted. The lifespan experiments were repeated at least three times with similar results, and the data for representative experiments are shown. The lifespan data were analyzed using the log-rank test and *p*-values for each individual experiment are shown. The *p*-value was calculated by comparing c-di-GMP-treated to the untreated worms.
[a]Results presented in Fig. 4a.
[b]Results presented in Fig. 4b. The worms were fed with heat-killed *E. coli* OP50.
[c]Results presented in Fig. 6a.
[d]Results presented in Fig. 6b.
[e]Results presented in Fig. 6c.
[f]Results presented in Fig. 7a.
[g]Results presented in Fig. 7b.
[h]Results presented in Fig. 7c.

We showed that c-di-GMP acts through two key stress response transcription factors in *C. elegans*, SKN-1 and HSF-1. SKN-1 is the downstream effector of the major immune-signaling p38 MAPK pathway and controls numerous genes involved in stress response and lifespan regulation[27]. HSF-1 is a multifaceted transcription factor involved in stress response, development, metabolism, as well as lifespan and immunity modulation[28–30]. The possible MAP kinases (PMK-1, JNK-1, and MPK-1) that regulate SKN-1 and HSF-1 activity and nucleus localization were investigated. PMK-1 from the p38 MAPK pathway and MPK-1 from the RTK-Ras-ERK pathway are required in the c-di-GMP-elicited signal transduction. It is known that both PMK-1 and MPK-1 can phosphorylate SKN-1 to modulate its function[32,33], while the kinase in *C. elegans* to phosphorylate HSF-1 is still not clear. Our data propose the possibility that PMK-1 or MPK-1 might be the upstream kinase for HSF-1. Of course, further studies are needed to address this interesting question.

Although we believe that we have successfully defined this new role for bacterial second messengers and their effect on eukaryotes, more work still needs to be done. Communications between bacteria and their hosts likely have played an indispensable role during their coevolution over millions of years.

Bacteria have evolved to release signaling molecules to improve their survival and spread by interfering with the host's behavior and immune response. In conjunction, the hosts have developed receptors for these signals to trigger immune responses to defend themselves. This has contributed to the symbiotic relationship that exists between bacteria and their hosts. Manipulation of such signaling can provide valuable insight to researchers looking to develop novel methods to treat infection and combat persistence. The better we understand this "language", the better we can start to control the conversation rather than reacting.

## Methods

**Plasmids, bacterial strains, and growth conditions**. The plasmid pBAD33 (pACYC184 *araC* P$_{araBAD}$, Cm$^r$)[44] was obtained from Coli Genetic Stock center, pAT1568 (pBAD33::NTF3*vieA*-His$_6$) was provided by Dr. Andrew Camilli (Tufts University)[18]. The *V. cholerae* strains used in this study were all derived from the wild-type C6706 strain (O1 serotype El Tor isolated from Peru)[45]. The *cqsA*$^-$ (*cqsA* deletion), *luxS*$^-$ (*luxS* deletion), *cqsA*$^-$/*luxS*$^-$ (*cqsA* and *luxS* double deletion) strains were obtained from Dr. Jay Zhu (University of Pennsylvania)[46–48]. *E. coli* OP50 came from the laboratory stocks. *E. coli* OP50 and *V. cholerae* were cultured in Lysogeny broth (LB) with 100 μg/mL of streptomycin. Chloramphenicol (5 μg/mL) was used for the selection of pBAD33 and pAT1568 in *V. cholerae*. L-arabinose was added at a final concentration of 0.2% to induce expression from P$_{araBAD}$.

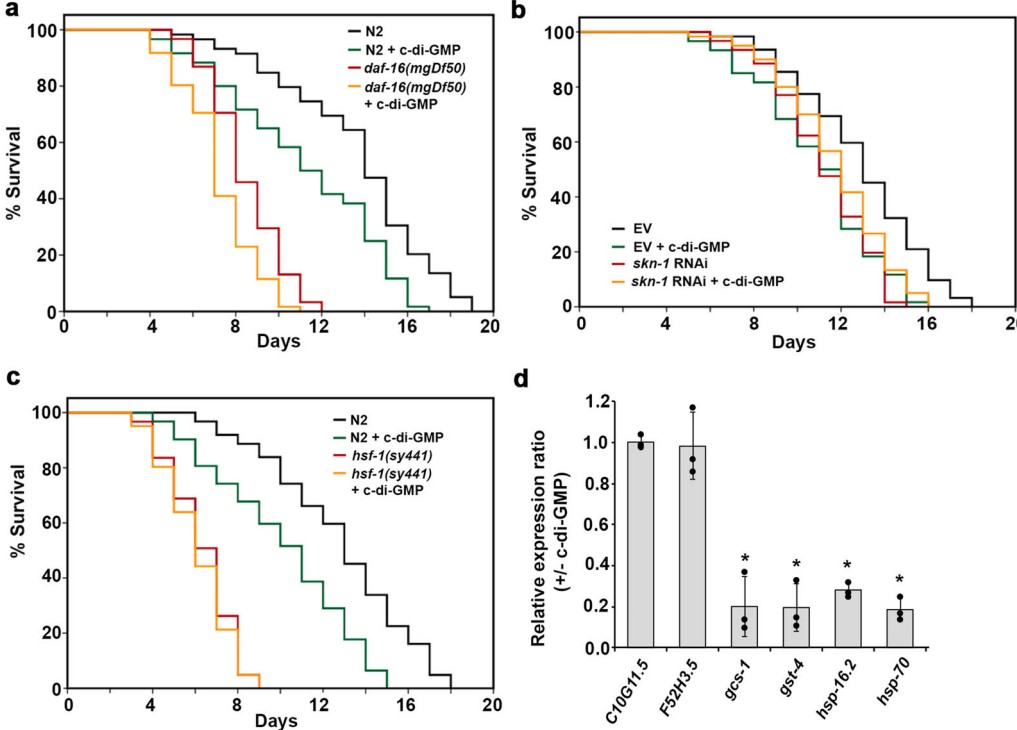

**Fig. 6 C-di-GMP acts through SKN-1 and HSF-1, but not DAF-16.** Lifespan of **a** *daf-16(mgDf50)*, **b** *skn-1* RNAi, and **c** *hsf-1(sy441)* worms when supplemented with 1 nM c-di-GMP. Each lifespan experiment was repeated in at least three independent trials with similar results. Quantitative data and statistical analyses for all trails are included in Table 1. EV stands for "empty vector pL4440". **d** qRT-PCR analysis of the representative target genes of DAF-16 (*C10G11.5* and *F52H3.5*), SKN-1 (*gcs-1* and *gst-4*), and HSF-1 (*hsp-16.2* and *hsp-70*), in response to 1 nM c-di-GMP (normalized to *act-1*). Results are the average of three independent experiments, and error bars are standard error of the mean representing 95% confidence intervals. Individual values are shown as black dots. *p* values were calculated by using the unpaired Student *T*-test (each gene vs. *act-1*). *$p < 0.05$.

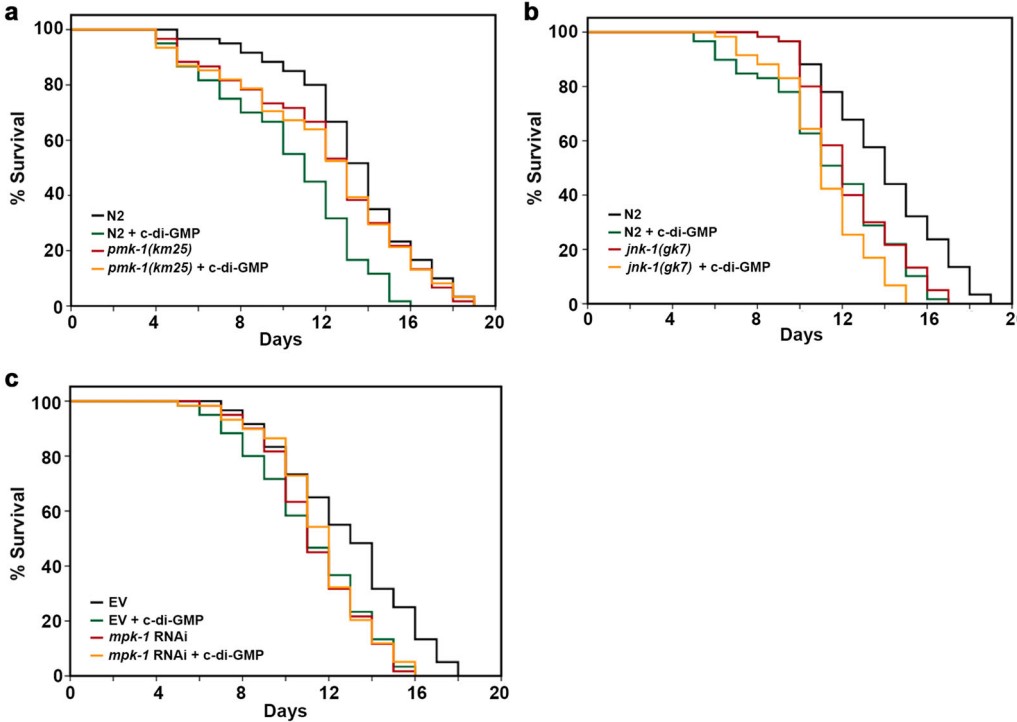

**Fig. 7 PMK-1 and MPK-1 are required in the c-di-GMP-elicited signal transduction pathway.** Lifespan of **a** *pmk-1(km25)*, **b** *jnk-1(gk7)*, and **c** *mpk-1* RNAi worms when supplemented with 1 nM c-di-GMP. Each lifespan experiment was repeated in at least three independent trials with similar results. Quantitative data and statistical analyses for all trails are included in Table 1. EV stands for "empty vector pL4440".

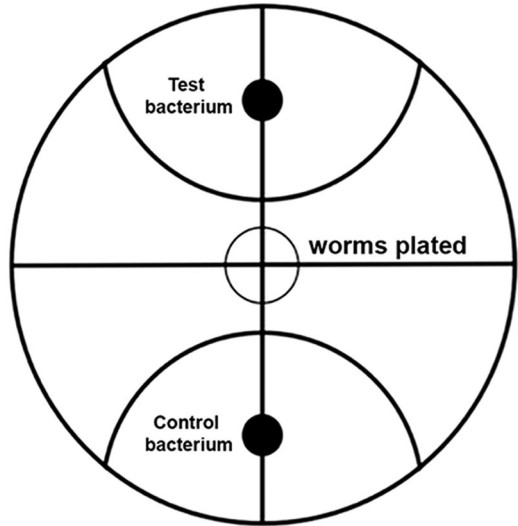

**Fig. 8 Diagram of the choice index assay.** The choice index assay plates were divided into quarters to reveal the center point in each plate. The center circle designates the initial position where the assay worms were placed. Overnight bacterial cultures were seeded onto each end of the plate 6 cm apart, and a 2 cm radius is drawn around each lawn. Worms present within the 2 cm radius of each bacterial lawn were counted after one and two hours.

**C. elegans strains and growth conditions.** All *C. elegans* strains were obtained from the Caenorhabditis Genetics Center, University of Minnesota, USA. Strains used in this study were: N2 Bristol (wild type), AU3 (*nsy-1(ag3)*), FK100/PR678 (*tax-2(ks10)/tax-4(p678)*), FG125 (*ocr-2(ak47), osm-9(ky10), ocr-1(ak46)*), FK311 (*ceh-36(ks86)*), PR674 (*che-1(p674)*), CX6161 (*inx-19(ky634)*, previously *nsy-5*), GR1370 (*daf-16 (mgDf50)*), PS3551 (*hsf-1(sy441)*), KU25 (*pmk-1(km25)*), and VC8 (*jnk-1(gk7)*),

All strains were maintained at their permissive temperature on nematode growth medium (NGM) seeded with *E. coli* OP50 feeding strain. NGM for maintenance was prepared via standard protocol in 60 mm plates. *E. coli* OP50 was dropped on the center of the plates the night prior to transfer of worms. Bacterial strains were dropped onto the center of the plates 2 h prior to worm transfer. NGM-FUDR 35 mm plates were used for lifespan assays. Fluorodeoxyuridine (FUDR) is an inhibitor of DNA synthesis, and at a concentration of 50 µg/mL it prevents the reproduction and development of progeny but doesn't interfere with the lifespan and the physiology of adult *C. elegans*[49,50]. Five times concentrated bacterial cultures were dropped in 100 µL aliquots onto the center of plates 2 h prior to worm transfer. NGM or NGM-FUDR plates with 1 nM c-di-GMP were created by added correct concentrations of stock c-di-GMP (1 mM) to warm NGM mixtures after autoclaving.

**Preparation of cyclic di-nucleotides.** Cyclic diadenosine monophosphate (c-di-AMP), cyclic diguanosine monophosphate (c-di-GMP), and cyclic adenosine monophosphate– guanine monophosphate (cGAMP) were purchased from BioLog Life Science Institute (catalog numbers are C 088, C 057, and C 117, respectively). Stock concentrations of these sodium salt compounds were made by dissolving in ddH$_2$O. The stock samples were stored in −20 °C freezer. Serial dilutions were made from frozen stocks to obtain desired concentration before experiments were conducted.

**Liquid chromatography-mass spectrophotometry (LC-MS) assay.** *V. cholerae* C6706 was grown shaking at 37 °C for 22 h in 100 mL of LB broth. The culture was centrifuged for 10 min at maximum speed to separate the supernatant and the cell pellet. The supernatant and the cell lysate were tested for the presence of CDNs, corresponding to the extracellular and intracellular concentrations, respectively. The supernatant was treated with final concentration of 10 mM EDTA for 10 min and then passed through 0.45-µ filter. The filtrate was freeze-dried, resuspended in 500 µL of sterile ddH$_2$O, and centrifuged in Amicon Ultra-0.5 mL centrifugal filter device to remove any DNA or protein molecules larger than 3 KDa. The sample was lyophilized and stored at room temperature before LC-MS assay. The cell pellet was suspended in 1 mL of 25 mM Tris-HCl + 10 mM EDTA buffer. Cells were then lysed by sonication at 20 W for 60 s (three 20 s pulses). Tricholoacetic acid at a final concentration of 12% was used to precipitate cellular macromolecules and centrifugation was used to separate the precipitate. The supernatant was freeze-dried, resuspended in 500 µL of sterile ddH$_2$O, and centrifuged in Amicon Ultra-0.5 mL centrifugal filter devices to remove any remaining DNA or protein

molecules larger than 3 KDa. The sample was lyophilized and stored at room temperature before LC-MS assay.

C-di-GMP and cGAMP in supernatant and cell pellet were quantified by HPLC with online tandem mass spectrometry (MS). The CDNs were resolved on a C18 column (150 × 2 mm, Prodigy$^{TM}$ 5 µm ODS-2 150 Å) (Phenomenx) using a gradient generated between 0.2% formic acid in water (A) and 0.2% formic acid in methanol (B) at a flow rate of 0.25 mL/min starting from 0% B over 2 min, then linearly to 20% B over 4 min, then to 100% B over 0.5 min, followed by this solution for 3 min, then back to 0% B. CDNs were analyzed on a Shimadzu 8050 triple-quadrupole mass spectrometer interfaced to a Shimadzu UHPLC multiplexing system using electrospray ionization in positive-ion mode with multiple reaction monitoring of parent and characteristic daughter ions *m/z* 675.10 → 136.10 for cGAMP and *m/z* 691.10 → 152.10 for c-di-GMP. Several other parent and characteristic daughter ions *m/z* 675.10 → 524.00, 675.10 → 330.05 and *m/z* 691.10 → 539.90, 691.10 → 248.05 were used as reference to confirm the defined cGAMP and c-di-GMP, respectively. The MS parameters were optimized by using their respective standards. Series concentrations of CDN standards were monitored by the above UHPLC/MS to determine retention times and regression equations to calculate the concentrations in supernatant and cell pellet.

**Choice index (CI) assays.** *C. elegans* were grown at 25 °C on *E. coli* OP50 under well-fed and un-crowded conditions. CI assays were performed on standard LB agar plates. The plates were divided in half to reveal the center point in each plate. Bacterial strains that were used for this experiment were grown in a shaking incubator overnight in the proper growth medium and temperature at approximately 120 rpm. Overnight cultures were then seeded onto each end of the plate 6 cm apart, and a 2 cm radius is drawn around each lawn (Fig. 8). The lawns are allowed to dry for 2 h before experiments were conducted. After the lawns were able to dry, 1 µL of 10 mM sodium azide (NaN$_3$) was dropped on the center of each lawn to paralyze *C. elegans* to make sure, they did not change bacterial lawns once one was initially chosen. Between 50 and 150 well-fed N2 worms were then placed in the center of the assay plate to begin the choice experiment. Worms present within the 2 cm radius of each bacterial lawn were counted after 1 and 2 h. Each assay was performed independently and in at least triplicate. Choice index was calculated as follows:

*Choice Index = (# of worms on test strain - # of worms on *E. coli* OP50) / Total # of worms.

*Positive values indicate preference towards the test strain, while negative values indicate preference towards *E. coli* OP50. If there were no preference, the equation would yield a value close to 0.

Average and standard error were calculated from three independent experiments.

Unpaired Student's *T*-test (between two groups) and one-way balanced ANOVA (for more than two groups) were used for statistical analysis. A *p*-value < 0.05 was accepted as statistically significant.

**C. elegans lifespan assay.** Lifespan assays were carried out at 25 °C. Worms were synchronized by transferring 20 gravid worms to 60 mm NGM plates seeded with *E. coli* OP50 2 days prior to the start of assays. Worms were allowed to lay eggs for 4 h, and then parent worms were removed from plates, leaving only synchronized eggs. Worms were then incubated until L4 stage. Overnight bacterial cultures were concentrated by centrifugation and removal of 50% of supernatant. Cultures were then resuspended in remaining medium. When necessary, *E. coli* OP50 was heat killed at this stage (1 h at 80 °C). Aliquots of 100 µL of *E. coli* OP50 were dropped on the center of NGM-FUDR 35 mm plates (with and without 1 nM c-di-GMP) and allowed to dry for 2 h. Worms in L4 stage were transferred to assay plates; 20 worms per plate. If the worms were transferred onto plates with heat-killed *E. coli* OP50, then prior to transferring to the assay plates they were allowed to roam on LB gentamycin (50 µg/mL) plates for 1 h to rid external live bacteria. Plates were subsequently checked daily. Dead worms were removed from the plates after being counted and recorded. The day of transfer was defined as day zero. Statistical analysis was carried out through SPSS software under the Kaplan–Meier lifespan analysis. The *p*-values were determined via log rank test, and a *p* < 0.05 was accepted as statistically significant.

**Gut colonization of C. elegans.** *E. coli* OP50 was grown overnight in LB broth with 100 µg/mL of streptomycin broth. Aliquots of 100 µL of *E. coli* OP50 was dropped on NGM with 1 nM c-di-GMP and control NGM plates. Five *C. elegans* N2 gravid worms were transferred onto each plate and incubated at 20 °C for 4 h. Synchronized egg populations were allowed to grow to L4 stage (~40 h), and then transferred to NGM-FUDR and NGM-FUDR with 1 nM c-di-GMP plates with 100 µL of respective overnight cultures at 2× concentration. This was designated as Day 0, and tests were conducted on Day 6 aged worms on each bacterial species +/− c-di-GMP. Ten worms were randomly chosen from each plate and transferred to LB gentamycin (50 µg/mL) plates for one hour to rid N2 worms of external bacteria. Worms were then transferred to Eppendorf tubes containing 400 mg of St. Si Carbide beads and 1 mL of M9. Tubes were vortexed for 1 min and then centrifuged at max rpm for 2 min. After, tubes were then quickly vortexed and serially diluted in M9 and plated on LB streptomycin (100 µg/mL) plates. Colony-forming

**Table 2 Oligonucleotides used in quantitative real-time PCR.**

| Gene | | Sequence of the oligonucleotide |
|---|---|---|
| act-1 | Forward | 5′-CCAGGAATTGCTGATCGTATGCAGAA-3′ |
| | Reverse | 5′-TGGAGAGGGGAAGCGAGGATAGA-3′ |
| C10G11.5 | Forward | 5′-AGTAGATTCAGCGACAGGGAGTGT-3′ |
| | Reverse | 5′-AAGCATCAGCAGCACATACTCGTG-3′ |
| C23G10.1 | Forward | 5′-CCATCCACTCTTGGTTGCTT-3′ |
| | Reverse | 5′-TCACGTGCTCCTTTTTCCTT-3′ |
| clec-46 | Forward | 5′-CTTCCTCGGTTCTTGCACTT-3′ |
| | Reverse | 5′-GCGGTTTCCAACAAAAACAC-3′ |
| clec-71 | Forward | 5′-TTGGCTGTTGTAGGCAATCA-3′ |
| | Reverse | 5′-TCACTGGGAATCCGTTATCC-3′ |
| col-41 | Forward | 5′-CACCAGGAACTCCAGGAAAC-3′ |
| | Reverse | 5′-GTGGGGTTCTGTCGTCTTGT-3′ |
| dct-5 | Forward | 5′-GCTGCAAAATGTGGAAATGA-3′ |
| | Reverse | 5′-AAGTTTTGGGCACAGTCCAG-3′ |
| dod-22 | Forward | 5′-TTGTTGGTCCCAAGTTCACA-3′ |
| | Reverse | 5′-AAGAACTTCGGCTGCTTCAG-3′ |
| F52H3.5 | Forward | 5′-GCATCCAGGCTATGAGCTCAGTTT-3′ |
| | Reverse | 5′-GCTTCATCCAACCGTACACCTTCT-3′ |
| fmo-2 | Forward | 5′-TGCTGTCATAGGAGCTGGTG-3′ |
| | Reverse | 5′-CATCTGACGCCTCAAAACAA-3′ |
| gcs-1 | Forward | 5′-GGAATGCCTTACGGAGGTC-3′ |
| | Reverse | 5′-CGATAGACATGTTTCATCCTTC-3′ |
| gst-4 | Forward | 5′-CTCTTGCTGAGCCAATCCGT-3′ |
| | Reverse | 5′-CTGGCCAAATGGAGTCGTTG-3′ |
| hsp-16.2 | Forward | 5′-TTGCCATCAATCTCAACGTC-3′ |
| | Reverse | 5′-CTTTCTTTGGCGCTTCAATC-3′ |
| hsp-70 | Forward | 5′-CGTTTCGAAGAACTGATCTATTCCGG-3′ |
| | Reverse | 5′-TTATCAACTTCCTAGGTCCTTGTGG-3′ |
| pqn-5 | Forward | 5′-GCTCAGCCACAACAAACTCA-3′ |
| | Reverse | 5′-CTGGCACTGTTGCTGACATT-3′ |

units (CFUs) were then determined after overnight incubation at 37 °C. Tests were conducted in three independent trials, between three independent populations and CFUs were averaged per N2 worm. Unpaired student's T-test (between two groups) was used for analysis, and a p-value < 0.05 was accepted as statistically significant.

**RNA interference (RNAi) of skn-1 and mpk-1**. The RNAi constructs targeting skn-1 was obtained from the C. elegans ORFeome RNAi library v1.1[51]. The RNAi constructs targeting mpk-1 was constructed by PCR amplification of exon 6 (forward primer: 5′-ATGGCTAGCCACGTGATTACATAGTTCAATGCCTGAT-3′ and reverse primer: 5′-TCG ATGTCGATACGATTATGTGGATTGAAAGTG-3′) and exon 7 (forward primer: 5′-ATAATCGTATCGACATCGAGCAAGCA-3′ and reverse primer: 5′-GGTCGACGGTATCGATACTAAACAGGATTCTGCCCT C-3′) of the mpk-1 gene and cloned between the NheI and SalI restriction sites of the pL4440-DEST vector. The skn-1 and mpk-1 RNAi plasmids were introduced into E. coli HT115(DE3) by transformation. The resulting transformants were inoculated in LB plus tetracycline (12.5 μg/mL) and carbenicillin (25 μg/mL) and grown at 37 °C with shaking for 8–12 h. Thereafter, isopropyl β-D-1-thiogalacto-pyranoside (IPTG) was added into the growth culture with a final concentration of 2 mM to induce the expression of the interfering RNA. Four hours after the induction, the bacteria were harvested by centrifugation. Ten-fold concentrated RNAi bacteria were seeded onto RNAi plates containing 25 μg/mL of carbenicillin. E. coli HT115(DE3) with the empty vector pL4440-DEST was cultured the same way and seeded onto the control plates. Synchronized worm populations were acquired by allowing 10–15 hermaphrodites to lay eggs on NGM plates seeded with E. coli HT115(DE3) with the RNAi plasmid or the empty vector (EV) for 4 h at permissive temperature, and then the parents were removed. Eggs left on the plate would reach L4 after around 40 h. The L4 worms were then transferred to corresponding RNAi or EV E. coli seeded NGM-FUDR plates (with and without 1 nM of c-di-GMP). Lifespan assays were conducted and analyzed as described above.

**Quantitative real-time PCR**. All RNA samples were prepared using RNAzol RT reagent (Molecular Research Center, INC.) and stored at −80 °C. Complementary DNA was synthesized using the BioRad iScript cDNA synthesis kit (Bio-Rad). For C. elegans immune response genes, gravid worms were synchronized on NGM plates seeded with E. coli OP50. Worms were allowed to lay eggs for 4 h and then gravid adults were killed leaving only synchronous egg population. Once nematodes reached young adult stage, M9 buffer was used to wash collected worms. Worm populations

were washed at least twice to remove any bacterium that was left in the media or on the worms. After this was completed, the population was split into 50 μg populations into separate sterile tubes. This is done to ensure an appropriate amount of RNA in each sample, as well as a high purity. One sample was treated with appropriate final concentration of 1 nM c-di-GMP, and the other sample served as a control.

qRT-PCR was performed using SensiFAST SYBR No-Rox Kit (Bioline) and the CFX96 real-time PCR detection system according to the manufacturer's suggested protocol (Bio-Rad). The qPCR conditions were: 95 °C for 3 min, followed by 40 cycles of 10 s at 95 °C and 30 s at 60 °C. Transcription levels of act-1 were used as internal controls to normalize the expression levels of target transcripts. Relative fold changes are then to be calculated using the comparative CT ($2^{−\Delta\Delta CT}$) method[52]. Cycle thresholds of amplification were determined by Light Cycler software (Bio-Rad). Each qPCR experiment was repeated three times using independent RNA preparations. The data were pooled and analyzed using unpaired Student's T-test (between each target gene and the control act-1), and a p-value < 0.05 was accepted as statistically significant. The primers used in this study are listed in Table 2.

**Statistics and reproducibility**. Each assay was conducted in at least three biological replicates. One biological replicate is defined as using one batch of synchronized C. elegans population and/or one batch of freshly prepared bacterial culture. For the choice index, LC-MS, gut colonization, and qPCR assays, average and standard error were calculated from data points collected from all three biological replicates. Unpaired Student's T-test (between two groups) and one-way balanced ANOVA (for more than two groups) were used for statistical analysis. A p-value < 0.05 was accepted as statistically significant. For the life span assays, statistical analysis was carried out through SPSS software under the Kaplan–Meier lifespan analysis for each replicate. The p-values were determined via log rank test, and a p < 0.05 was accepted as statistically significant. Analyses from all three biological replicates were compiled and similar results were obtained.

**Reporting summary**. Further information on research design is available in the Nature Research Reporting Summary linked to this article.

## Data availability
The authors declare that all data supporting the findings of this study are available within this article, Supplementary material, and Supplementary data set. Source data underlying the graphs in figures are provided in Supplementary data set.

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

## Acknowledgements
We would like to thank Dr. Jay (Jun) Zhu for generously providing all the *V. cholerae* strains and Dr. Andrew Camilli for providing the pAT1568 plasmid. This work was supported by the Clemson University Creative Inquiry fund and the Clemson SEED grant to M.C. and the National Institutes of Health and the Office of Dietary Supplements, R01HL130819, to Z.W.

## Author contributions
M.C. conceived the project; J.A. and M.C. designed the experiments; J.A., Y.D., and Z.W. performed the experiments; M.C. wrote the paper with input from all authors.

## Competing interests
The authors declare no competing interests.
