## [Peer Review File · Communications Biology]

Reviewers' comments:

Reviewer #1 (Remarks to the Author):

In this manuscript, the authors provide mechanistic insight on how the bacterial secondary messenger c-di-GMP affects the eukaryotic host *C. elegans*. Authors did a very good job on presenting their findings in a very clear way, making the manuscript clear and easy to follow.

I have some small experimental questions.

1) Lifespan experiments, shown in Figure 5, demonstrate N2 lifespan shortening in the presence of c-di-GMP. This shortening is possibly due to the increased bacterial colonization because of the inhibition of host immunity. However, to unambiguously confirm that, lifespan experiments on heat- (or otherwise) killed *E. coli* need to be performed. People used to trust heat-killed *E. coli* experiments less because worms grown on it were small and sickly. However, a recent paper from Min Han's lab showed that addition of vitamin B2 to heat-killed bacteria allows normal worm growth (PMID: 28569665).

2) It looks like the basal level of immune genes were measured via qPCR in Figure 6. Usually, immune genes are not expressed particularly highly in the absence of a pathogen. Please expose worm to live pathogen with known immune response (*P. aeruginosa* may be a good choice, as both, PMK-1 and SKN-1 are involved in defense), and measure immune genes +/- c-di-GMP supplementation. Also, a number toxic molecules can activate immune response on their own (Cry5B, ToxA, etc). Alternatively, these toxins can be added to live bacteria fed to worms. These can then be split with and without c-di-GMP.

3) Fig 7. shows decreased expression of SKN-1 and HSF-1 effectors. Please include two DAF-16 effectors as a negative control.

4) It looks like c-di-GMP acts rather upstream. Can this be proven using epistasis experiments? E.g. mutation or RNAi knockdown of the phosphatase VHP-1 will artificially activate PMK-1. What would the effect of c-di-GMP on the lifespan of *vhp-1* worms be? Or what is the effect of c-di-GMP on the expression of PMK-1 targets in a *vhp-1* background?

5) How does c-di-GMP inhibit PMK-1? Is there decrease in its transcription? Does it affect the protein level or its phosphorylation? To answer these questions, there is a PMK-1::GFP reporter. Mammalian anti-phospho-p38 antibodies are known to work for *C. elegans* PMK-1 as well.

With regard to the discussion, it seems like the authors mainly restate their own findings. In the last 5-10 years there have been more than 500 papers on c-di-GMP, and a number of them describe effects of bacterial secondary messengers, etc. on eukaryotes (including *C. elegans* or mammalian cells). The discussion would be more helpful if the authors widened their scope to papers from other labs and placed their findings in the larger context of what is known in the field in general.

Reviewer #2 (Remarks to the Author):

The authors provide interesting data to suggest that the intracellular signaling molecule c-di-GMP could act as a chemoattractant. This study could add a new twist to the emerging roles of cyclic dinucleotides in inter kingdom interactions. But some rigorous experiments need to be done to improve the study:

1) It is important to obtain dose responses for many of the experiments (such as a few of the

survival studies, the signaling aspect etc.). I suggest that the authors also profile 10 nM and 100 nM. If chemoattraction is suggested, it is critical to show effects are different concentrations.

2) It is likely that in real systems, other guanine nucleotides will also be available. Is this effect specific for c-di-GMP or is c-di-GMP signaling through general guanine nucleotide pathways? To adequately address this, I suggest that the authors compete c-di-GMP with pGpG (the degradation product of c-di-GMP), cGMP, cGMP and GTP to determine if antagonism or synergism is seen. Also important to profile the effects of these alone in their system for comparison.

In this study, the authors show that *C. elegans* is attracted to a compound produced by the pathogen *Vibrio cholerae* (c-di-GMP), and they elucidate how it is sensed by the worms and how the compound leads to weakened immunity in the worms.

The study investigates an interesting interaction between a host and a pathogen, and I find the results to be intriguing and the manuscript well-written overall.

My only major complaints would be the following:

- a) The author state in the discussion that they showed that c-di-GMP “promotes bacterial invasion and establishment in their hosts”, however all gut colonization/lifespan/immunity experiments as far as can tell were done with c-di-GMP treated worms and the food strain *E. coli*, not the actual pathogen *Vibrio*. The lines of experiments I can see is: a) Worms are attracted to *Vibrio* because of c-di-GMP made by *Vibrio*, b) c-di-GMP-treated worms get colonized more by *E. coli*, and live less long (when exposed to *E. coli*). Where are the colonization experiments for *Vibrio* here? Ideally with a c-di-GMP null mutant as comparison. I realize that this mutant will have a range of pleiotropic phenotypes (other than not producing c-di-GMP), so there are likely technical limitations to this. In this case, the discussion should reflect this, because it would be an obvious experiment to do, and it would help the reader if this gap was addressed in writing.
- b) I am not convinced about the relevance of this host-pathogen relationship in nature: would *Vibrio cholerae* and *C. elegans* ever meet? If yes, where (add to discussion)? If not, what can we learn from this (playing devil’s advocate here)? If the link to natural interactions could be the fact that c-di-GMP is conserved among bacteria, including relevant *C. elegans* pathogens, this should be discussed.
- c) I am not convinced that secreting c-di-GMP is a specific adaptation for *Vibrio* to gain access to hosts / to make infections easier. If c-di-GMP is so widely conserved among Gram-negative bacteria, what about the natural members of the *C. elegans* microbiome? Would they not produce/secrete c-di-GMP? If *C. elegans* feeds on bacteria in nature and takes up a lot of dead bacteria, would that not lead to high levels of free c-di-GMP in the gut? What are c-di-GMP levels in the gut of a normal, naturally-fed worm that contains more than just one *E. coli* food species (OP50)? And what would then change if *Vibrio* invaded this natural community?
- d) The figures should be improved by adhering to best practises in data visualization. Barcharts with error bars hide the underlying raw datapoints and can obscure differences in data distribution (see e.g. Figure 1 here: <https://doi.org/10.1371/journal.pbio.1002128>) To avoid this, barcharts can easily be overlaid with the raw datapoints for full transparency.

In summary, I think the study is well-done and no further experiments are needed, but the discussion is mostly focussed on the mechanism, and it could benefit from a wider discussion on what the effect of this chemical on worm behaviour and physiology could mean in the bigger context of host-pathogen interactions and pathogen ecology, beyond just *Vibrio*.

Minor comments:

1. Methods (Choice Index): Why were the worms grown at 25C rather than at 20C?
2. Methods (Choice Index): The authors say they paralysed the worms once they touch the spot, but what about if they have to experience both lawns first to make a decision?
3. Methods Lifespan: Dead worms were counted and recorded, but also removed?

4. The grinder section (beginning of page 17: starting with “Younger worms ...”, ending with “... and proliferate”) needs referencing.
5. Discussion: Sentence starting with “emerging evidence...”, I would expect a couple references at the end of this sentence.
6. qPCR data: please add all genes analyzed and their expression values as supplementary data to increase transparency of the analysis. As it says in the data availability statement: “All data generated or analyzed in this study are included in this article and the supplementary data file.” I could not find the qPCR data here.
7. Typos and phrasing:
 - page 3 potassium ion channels
 - page 8 bacterial strains
 - page 15 “on longer” → no longer
 - page 19 “no long reduce” → no longer
 - page 20 “enter the eukaryotic kingdom” → enter eukaryotic cells
8. Needs referencing: page 20, the sentence ending with “ ... and mate.”

We appreciate the reviewers' insightful suggestions and comments, all of which are very helpful to the improvement of our manuscript. Below, we have listed our point-by-point responses.

Reviewer #1:

1) Lifespan experiments, shown in Figure 5, demonstrate N2 lifespan shortening in the presence of c-di-GMP. This shortening is possibly due to the increased bacterial colonization because of the inhibition of host immunity. However, to unambiguously confirm that, lifespan experiments on heat- (or otherwise) killed *E. coli* need to be performed. People used to trust heat-killed *E. coli* experiments less because worms grown on it were small and sickly. However, a recent paper from Min Han's lab showed that addition of vitamin B2 to heat-killed bacteria allows normal worm growth (PMID: 28569665).

Our response: Thank you for your suggestion. We agree with the reviewer here in that we believe conducting these experiments would indeed be helpful and would better explain the workings of this signaling molecule. We expect no changes to be seen between the control population and the population exposed to c-di-GMP if the worms were fed with dead bacteria.

We have conducted the experiment as suggested using heat-killed *E. coli* OP50 to feed the N2 worms. The result is included in the new Fig 5B and Table 2. We added the following sentence, "We also found that the presence of live bacteria is requisite, as c-di-GMP only caused a slight decrease in the lifespan when the N2 worms were fed on heat-killed *E. coli* OP50 (Fig 5B and Table 2)." to the manuscript line 397-399.

2) It looks like the basal level of immune genes were measured via qPCR in Figure 6. Usually, immune genes are not expressed particularly highly in the absence of a pathogen. Please expose worm to live pathogen with known immune response (*P. aeruginosa* may be a good choice, as both, PMK-1 and SKN-1 are involved in defense), and measure immune genes +/- c-di-GMP supplementation. Also, a number toxic molecules can activate immune response on their own (Cry5B, ToxA, etc). Alternatively, these toxins can be added to live bacteria fed to worms. These can then be split with and without c-di-GMP.

Our response: Thank you for your suggestion. Regarding this study and the point we are trying to make is that under non-infectious conditions (*i.e.*, before directly contacting with the bacterial pathogen), this very initial attraction towards a bacterial lawn with this signaling molecule present can suppress immunity in the very early stages, so that the host (*C. elegans*) becomes vulnerable to the upcoming pathogen. After this attraction, the worms have already chosen this bacterium and will most likely consume it. In regard to the candidate pathogens the reviewer mentioned, we believe this initial suppression of immunity will not last long, and the immune response would be initiated shortly thereafter. Since we noticed the initial attraction towards this signal, and also showed the production of this signal in *V. cholerae* is vital for attractive behavior to this bacterium, we set up gene expression analysis as following: synchronize *C. elegans* populations with *E. coli* OP50 as a food source, when the synchronized population reaches L4 stage the worms are sampled, and half of the population is then exposed to 1 nM c-di-GMP for just 10 min.

These pathogens, and most Gm⁻ pathogens, have a number of genes responsible for the production of c-di-GMP. *P. aeruginosa* and *V. cholerae* possess some of the highest numbers of

genes responsible for controlling the production of c-di-GMP that we know of to date. Production of c-di-GMP and its signaling controls many genes involved in virulence and biofilm formation in bacteria. *E. coli* OP50 does not possess these domains, as far as we know. If we tried to set up similar experiments utilizing these pathogens, we would run into several problems. First, the worms would have to be synchronized and grown to the L4 stage on *E. coli* OP50 and then transferred onto plates with the pathogens. We would have to determine the amount of time these worms spend on these pathogens that would elicit a significant immune response from the genes tested. At this point, as we have shown through LC-MS results, these populations will most likely have been exposed to c-di-GMP at high concentrations. Therefore, treating with 1 nM c-di-GMP after this point might not have any effect at all as the signal transduction pathway has already been employed. As suggested by the reviewer, we could use some known toxins in combination with c-di-GMP to treat *C. elegans* and then measure the changes of the immune genes. However, this would deviate from our original intention, which was mentioned in the last paragraph.

3) Fig 7. shows decreased expression of SKN-1 and HSF-1 effectors. Please include two DAF-16 effectors as a negative control.

Our response: We picked two DAF-16 target genes (*C10G11.5* and *F52H3.5*) (Ref: S. S. Lee *et al.*, DAF-16 Target Genes That Control *C. elegans* Life-Span and Metabolism (2003) *Science* 300, 644-47.) and measured their expression in the presence and absence of c-di-GMP. The result is included in Fig 7D. As expected, the expression of these two DAF-16 targets did not change upon c-di-GMP exposure. We edited line 445-449, “Next, two representative target genes of DAF-16 (*C10G11.5* and *F52H3.5*), two of SKN-1 (*gcs-1* and *gst-4*), and two of HSF-1 (*hsp-16.2* and *hsp-70*) were measured for their expression levels in the presence and absence of c-di-GMP. Fig 7D shows that c-di-GMP triggered down-regulation of the SKN-1 and HSF-1 target genes but not the DAF-16 targets.” The primers for these genes were added to Table 1.

4) It looks like c-di-GMP acts rather upstream. Can this be proven using epistasis experiments? E.g. mutation or RNAi knockdown of the phosphatase VHP-1 will artificially activate PMK-1. What would the effect of c-di-GMP on the lifespan of *vhp-1* worms be? Or what is the effect of c-di-GMP on the expression of PMK-1 targets in a *vhp-1* background?

Our response: Thank you for the suggestion. We believe that these experiments would be great to conduct to further elucidate PMK-1 and its role in c-di-GMP signal transduction. We were planning on ordering the strain JT366 that harbors a mutation in *vhp-1*. This would result in increased activity of PMK-1. If we conducted lifespan assays with this mutant strain with and without c-di-GMP and compared it to the results we obtained with the *pmk-1* strain (where we saw decreased in lifespan), if our theory holds and this kinase is involved then we would expect no difference in lifespan with the *vhp-1* strain. We could also look at gene expression levels +/- c-di-GMP of this *vhp-1* strain (*gst-4* and *gcs-1*).

Due to the impact of COVID-19, the Caenorhabditis Genetics Center (CGC), where we usually request *C. elegans* strains, stopped taking new orders since March 23. Although the center has resumed partial operations since June 2, the order processing time is much longer than usual. We are interested in elucidating the upstream pathways involved in c-di-GMP signaling, but we feel this suggested experiment is not indispensable for the conclusion of this manuscript. We hope the reviewer could accept our apology for not being able to perform this experiment under the current situation.

5) How does c-di-GMP inhibit PMK-1? Is there decrease in its transcription? Does it affect the protein level or its phosphorylation? To answer these questions, there is a PMK-1::GFP reporter. Mammalian anti-phospho-p38 antibodies are known to work for C. elegans PMK-1 as well.

Our response: In this study, we showed that two MAPKs, PMK-1 and MPK-1, are required in the c-di-GMP-elicited signal transduction. It is intriguing to elucidate how PMK-1 and MPK-1 are affected by c-di-GMP, as well as the upstream pathway(s).

In an earlier study (“Cranberry extract standardized for proanthocyanidins promotes the immune response of *Caenorhabditis elegans* to *Vibrio cholerae* through the p38 MAPK pathway and HSF-1.” (2014) PLoS ONE 9(7): e103290.), we used the anti-phospho-p38 MAPK mouse monoclonal antibody (28B10) from Cell Signaling and successfully detected phosphorylated PMK-1. We tried the same antibody but couldn’t get any signals this time. We also requested some samples from ThermoFisher, abcam, and Santa Cruz Biotechnology, but none of these worked. Also, as mentioned above, ordering strains from CGC takes a longer time. We are interested in elucidating the upstream pathways involved in c-di-GMP signaling, but we feel this suggested experiment is not indispensable for the conclusion of this manuscript. We hope the reviewer could accept our apology for not being able to perform these experiments under the current situation.

With regard to the discussion, it seems like the authors mainly restate their own findings. In the last 5-10 years there have been more than 500 papers on c-di-GMP, and a number of them describe effects of bacterial secondary messengers, etc. on eukaryotes (including C. elegans or mammalian cells). The discussion would be more helpful if the authors widened their scope to papers from other labs and placed their findings in the larger context of what is known in the field in general.

Our response: Thank you for your comments. We have substantially revised the discussion. For example, we discussed the interaction of *C. elegans* and bacteria in their natural environment and the broad implication of our findings (line 482-502). We also included more discussion on the STING research field (line 507-528).

Reviewer #2:

The authors provide interesting data to suggest that the intracellular signaling molecule c-di-GMP could act as a chemoattractant. This study could add a new twist to the emerging roles of cyclic dinucleotides in inter kingdom interactions. But some rigorous experiments need to be done to improve the study:

1) It is important to obtain dose responses for many of the experiments (such as a few of the survival studies, the signaling aspect etc.). I suggest that the authors also profile 10 nM and 100 nM. If chemoattraction is suggested, it is critical to show effects are different concentrations.

Our response: Thank you for your suggestion. In the choice index assays, we did test 5 nM. Our results showed that at 5 nM, the choice index is almost zero for both c-di-GMP and cGAMP (see the figure below). However, we don’t think this is a valid result. The way we calculate CI is by counting worms present within the 2 cm radius of each bacterial lawn (Fig 1) and then using the formula:

*Choice Index = (# of worms on Test strain - # of worms on *E. coli* OP50) / Total # of worms. The signaling molecule (c-di-GMP or cGAMP) diffuses from the dropping spot into the agar in decreasing amounts the further it is away from the dropping spot. At 5 nM, c-di-GMP or cGAMP can diffuse further away and maybe reach the center of the plate, so that the worms don't need to move toward the signal (instead, the signal reaches the worms). If we were to test 5 nM or higher concentrations, we would have to increase the distance between the signal spot and the worms (i.e., to use a much larger plate). We expect to see a similar attraction as that observed with 1 nM.

Regarding the survival assay with higher concentrations of c-di-GMP, this study and the point we are trying to make is that *C. elegans* can detect bacterial food from far away by sensing c-di-GMP (at a low concentration) released by bacteria in the environment, and this very initial attraction towards a bacterial lawn with this signaling molecule present can suppress immunity in the very early stages so that the host (*C. elegans*) becomes vulnerable to the upcoming bacteria. From the LC-MS assay, we detected ~ 20 nM of c-di-GMP from the supernatant of *V. cholerae* overnight culture. We believe that the concentration of c-di-GMP in *C. elegans*' natural environment would be much lower. Therefore, we did not perform the survival assay with higher concentrations of c-di-GMP. We also considerably revised the discussion to reflect this concern.

2) It is likely that in real systems, other guanine nucleotides will also be available. Is this effect specific for c-di-GMP or is c-di-GMP signaling through general guanine nucleotide pathways? To adequately address this, I suggest that the authors compete c-di-GMP with pGpG (the degradation product of c-di-GMP), cGMP, cGMP and GTP to determine if antagonism or synergism is seen. Also important to profile the effects of these alone in their system for comparison.

Our response: Thank you for the advice. In Fig 3A, we showed that c-di-GMP and cGAMP have chemoattraction, while c-di-AMP has a repulsive effect. Since c-di-GMP and cGAMP have the common GMP part, we decided to test GMP and the similar molecule, cGMP (which is also a second messenger), as suggested. Our results showed that neither cGMP nor GMP could cause any chemoattraction or repulsion. The data are included in Fig 3A. The following “We also tested similar signaling molecules, GMP and cGMP, and showed that neither cGMP nor GMP could cause any chemoattraction or repulsion at concentrations similar to what elicited a response from c-di-GMP and cGAMP (Fig 3A).” was added in the manuscript (line 329-332). In Fig 3C, we showed that when the phosphodiesterase (which hydrolyzes c-di-GMP to pGpG) was overexpressed, there was no attraction or repulsion. This means that pGpG does not have such effect. Therefore, we believe the chemoattraction effect is specific for c-di-GMP and cGAMP. In Fig 3B, we showed the detection of c-di-GMP (not cGAMP) in both the cell lysate and the

supernatant, and in Fig 5A, we showed that only c-di-GMP caused shortened lifespan. Hence, we only focused on c-di-GMP in this study.

Because cGMP, GMP, and pGpG did not show chemoattraction or repulsion, these molecules should not have any antagonistic or synergistic effect. Therefore, the competition assay (c-di-GMP vs. cGMP, c-di-GMP vs. GMP, and c-di-GMP vs. pGpG) would not provide any new conclusions or add on to the existing ones.

Reviewer #3:

In this study, the authors show that *C. elegans* is attracted to a compound produced by the pathogen *Vibrio cholerae* (c-di-GMP), and they elucidate how it is sensed by the worms and how the compound leads to weakened immunity in the worms. The study investigates an interesting interaction between a host and a pathogen, and I find the results to be intriguing and the manuscript well-written overall.

My only major complaints would be the following:

a) The author state in the discussion that they showed that c-di-GMP “promotes bacterial invasion and establishment in their hosts”, however all gut colonization/lifespan/immunity experiments as far as can tell were done with c-di-GMP treated worms and the food strain *E. coli*, not the actual pathogen *Vibrio*. The lines of experiments I can see is: a) Worms are attracted to *Vibrio* because of c-di-GMP made by *Vibrio*, b) c-di-GMP-treated worms get colonized more by *E. coli*, and live less long (when exposed to *E. coli*). Where are the colonization experiments for *Vibrio* here? Ideally with a c-di-GMP null mutant as comparison. I realize that this mutant will have a range of pleiotropic phenotypes (other than not producing c-di-GMP), so there are likely technical limitations to this. In this case, the discussion should reflect this, because it would be an obvious experiment to do, and it would help the reader if this gap was addressed in writing.

Our response: Thank you for the suggestion. Ideally, *V. cholerae* wild-type/pBAD33 (empty vector) and wt/pAT1568 (overexpression of PDE when induced by L-arabinose → decreasing in intracellular c-di-GMP) could be used in the suggested experiments. However, c-di-GMP also regulates *V. cholerae* biofilm formation and virulence. Feeding *C. elegans* with these two different strains will elicit different immunological responses and affect lifespan. Therefore, gut colonization and lifespan cannot be simply measured due to this limitation. We have extensively revised the discussion. We hope that we have addressed the concern.

b) I am not convinced about the relevance of this host-pathogen relationship in nature: would *Vibrio cholerae* and *C. elegans* ever meet? If yes, where (add to discussion)? If not, what can we learn from this (playing devil’s advocate here)? If the link to natural interactions could be the fact that c-di-GMP is conserved among bacteria, including relevant *C. elegans* pathogens, this should be discussed.

Our response: Thank you for the suggestion. We have substantially revised the discussion and added a paragraph (line 482-502) to address the broad implication of our findings.

c) I am not convinced that secreting c-di-GMP is a specific adaptation for *Vibrio* to gain access to hosts / to make infections easier. If c-di-GMP is so widely conserved among Gram-negative bacteria, what about the natural members of the *C. elegans* microbiome? Would they not produce/secrete c-di-GMP? If *C. elegans* feeds on bacteria in nature and takes up a lot of dead

bacteria, would that not lead to high levels of free c-di-GMP in the gut? What are c-di-GMP levels in the gut of a normal, naturally-fed worm that contains more than just one *E. coli* food species (OP50)? And what would then change if *Vibrio* invaded this natural community?

Our response: Thank you for the suggestion. We included more discussion on the interaction of *C. elegans* and bacteria in their natural environment (line 482-502).

d) The figures should be improved by adhering to best practises in data visualization. Barcharts with error bars hide the underlying raw datapoints and can obscure differences in data distribution (see e.g. Figure 1 here: <https://doi.org/10.1371/journal.pbio.1002128>) To avoid this, barcharts can easily be overlaid with the raw datapoints for full transparency.

Our response: Thank you for your advice. We have remade all the bar graphs in Figures 2 – 7. All raw data points have been included in the graph.

In summary, I think the study is well-done and no further experiments are needed, but the discussion is mostly focused on the mechanism, and it could benefit from a wider discussion on what the effect of this chemical on worm behavior and physiology could mean in the bigger context of host-pathogen interactions and pathogen ecology, beyond just *Vibrio*.

Minor comments:

1. Methods (Choice Index): Why were the worms grown at 25C rather than at 20C?

R: Both 20°C and 25°C can be used. *C. elegans* normally lives longer at 20°C. We have been using 25°C in all experiments and thus want to keep it consistent.

2. Methods (Choice Index): The authors say they paralysed the worms once they touch the spot, but what about if they have to experience both lawns first to make a decision?

R: The reason we used NaN₃ is to follow the same protocol used by Beale et al. (Ref: 14) and Werner et al. (Ref: 15). During the early stage of this study, we tested the CI without adding NaN₃ and obtained similar results. We also performed a learning test to see whether *C. elegans* would learn to avoid *V. cholerae* if being pre-exposed to this pathogen for 1 or 2 hours. But no aversive learning was observed in these worms.

3. Methods Lifespan: Dead worms were counted and recorded, but also removed?

R: Yes, the dead worms were removed from the lifespan assay plates after being counted and recorded.

4. The grinder section (beginning of page 17: starting with “Younger worms ...”, ending with “... and proliferate”) needs referencing.

R: This sentence is followed by two sentences, and the references were given at the end of the third one. (Ref: 29, 30)

5. Discussion: Sentence starting with “emerging evidence...”, I would expect a couple references at the end of this sentence.

R: This sentence is actually followed by three sentences that provide examples and references (Ref: 10, 14, 15, and 43).

6. qPCR data: please add all genes analyzed and their expression values as supplementary data to increase transparency of the analysis. As it says in the data availability statement: “All data generated or analyzed in this study are included in this article and the supplementary data file.” I could not find the qPCR data here.

R: We have remade all the bar graphs in Figures 2 – 7. All raw datapoints have been included in the graph.

7. Typos and phrasing:

o page 3 potassium ion channels

o page 8 bacterial strains

o page 15 “on longer” no longer

o page 19 “no long reduce” no longer

o page 20 “enter the eukaryotic kingdom” enter eukaryotic cells

R: We have fixed all the typos except the last one, which we meant to say that bacterial CDNs enter the eukaryotic kingdom.

8. Needs referencing: page 20, the sentence ending with “... and mate.”

R: We have added the reference to this sentence (Ref: 28).

My only major complaints would be the following:

a) The author state in the discussion that they showed that c-di-GMP “promotes bacterial invasion and establishment in their hosts”, however all gut colonization/lifespan/immunity experiments as far as can tell were done with c-di-GMP treated worms and the food strain E. coli, not the actual pathogen Vibrio. The lines of experiments I can see is: a) Worms are attracted to Vibrio because of c-di-GMP made by Vibrio, b) c-di-GMP-treated worms get colonized more by E. coli, and live less long (when exposed to E. coli). Where are the colonization experiments for Vibrio here? Ideally with a c-di-GMP null mutant as comparison. I realize that this mutant will have a range of pleiotropic phenotypes (other than not producing c-di-GMP), so there are likely technical limitations to this. In this case, the discussion should reflect this, because it would be an obvious experiment to do, and it would help the reader if this gap was addressed in writing.

Our response: Thank you for the suggestion. Ideally, *V. cholerae* wild-type/pBAD33 (empty vector) and wt/pAT1568 (overexpression of PDE when induced by L-arabinose → decreasing in intracellular c-di-GMP) could be used in the suggested experiments. However, c-di-GMP also regulates *V. cholerae* biofilm formation and virulence. Feeding *C. elegans* with these two different strains will elicit different immunological responses and affect lifespan. Therefore, gut colonization and lifespan cannot be simply measured due to this limitation. We have extensively revised the discussion. We hope that we have addressed the concern.

Reviewer: Regarding the suggested experiments – ok. Regarding the revised discussion, see my comments below (response to comment C).

b) I am not convinced about the relevance of this host-pathogen relationship in nature: would Vibrio cholerae and C. elegans ever meet? If yes, where (add to discussion)? If not, what can we learn from this (playing devil’s advocate here)? If the link to natural interactions could be the fact that c-di-GMP is conserved among bacteria, including relevant C. elegans pathogens, this should be discussed.

Our response: Thank you for the suggestion. We have substantially revised the discussion and added a paragraph (line 482-502) to address the broad implication of our findings.

Reviewer: Regarding the revised discussion, see my comments below (response to comment C).

c) I am not convinced that secreting c-di-GMP is a specific adaptation for Vibrio to gain access to hosts / to make infections easier. If c-di-GMP is so widely conserved among Gram-negative bacteria, what about the natural members of the C. elegans microbiome? Would they not

produce/secrete c-di-GMP? If *C. elegans* feeds on bacteria in nature and takes up a lot of dead bacteria, would that not lead to high levels of free c-di-GMP in the gut? What are c-di-GMP levels in the gut of a normal, naturally-fed worm that contains more than just one *E. coli* food species (OP50)? And what would then change if *Vibrio* invaded this natural community?

Our response: Thank you for the suggestion. We included more discussion on the interaction of *C. elegans* and bacteria in their natural environment (line 482-502).

Reviewer: The revised discussion discusses the findings more broadly, and overall improves the manuscript substantially. The only remaining statement I still have issues with is line 491-493:

*“This [attraction of worms and infection] would allow *V. cholerae* to colonize the gut of a new host, cause infection, and finally be released into the environment to start the cycle again.”*

Since *V. cholerae* in nature is (to the best of my knowledge) exclusively associated with zooplankton, shellfish and other water-dwelling organisms, and the findings do seem to be specific to a soil-dwelling nematode encountering bacteria, I don't think it is appropriate to speculate about the role of c-di-GMP in the *Vibrio* lifecycle here, since the study does not monitor the response of a natural *Vibrio* host to c-di-GMP. The statement is also not necessary in discussing the main point of the story (c-di-GMP acting as a cue to attract bacterivore nematodes to a food source), so I suggest this statement be removed.

d) The figures should be improved by adhering to best practises in data visualization. Barcharts with error bars hide the underlying raw datapoints and can obscure differences in data distribution (see e.g. Figure 1 here: <https://doi.org/10.1371/journal.pbio.1002128>) To avoid this, barcharts can easily be overlaid with the raw datapoints for full transparency.

Our response: Thank you for your advice. We have remade all the bar graphs in Figures 2 – 7.

All raw data points have been included in the graph.

Reviewer: Ok great. I appreciate that the authors have remade the graphs, and also that there is now a supplementary data table containing the data points used to produce the graphs.

In summary, I think the study is well-done and no further experiments are needed, but the discussion is mostly focused on the mechanism, and it could benefit from a wider discussion on what the effect of this chemical on worm behavior and physiology could mean in the bigger context of host-pathogen interactions and pathogen ecology, beyond just *Vibrio*.

Minor comments:

1. Methods (Choice Index): Why were the worms grown at 25C rather than at 20C?

R: Both 20oC and 25oC can be used. *C. elegans* normally lives longer at 20oC. We have been using 25oC in all experiments and thus want to keep it consistent.

Reviewer: Ok.

2. Methods (Choice Index): The authors say they paralysed the worms once they touch the spot, but what about if they have to experience both lawns first to make a decision?

R: The reason we used NaN3 is to follow the same protocol used by Beale et al. (Ref: 14) and Werner et al. (Ref: 15). During the early stage of this study, we tested the CI without adding NaN3 and obtained similar results. We also performed a learning test to see whether *C. elegans* would learn to avoid *V. cholerae* if being pre-exposed to this pathogen for 1 or 2 hours. But no aversive learning was observed in these worms.

Reviewer: Ok.

3. Methods Lifespan: Dead worms were counted and recorded, but also removed?

R: Yes, the dead worms were removed from the lifespan assay plates after being counted and recorded.

Reviewer: Ok. This should be added to the methods description.

4. The grinder section (beginning of page 17: starting with “Younger worms ...”, ending with “... and proliferate”) needs referencing.

R: This sentence is followed by two sentences, and the references were given at the end of the third one. (Ref: 29, 30)

Reviewer: To me as a reader, this reads like refs 29+30 only support the claims on bacterial accumulation and gut immunity/lifespan, and not the claims from the sentences before. Consider adding references after each sentence (just a suggestion).

5. Discussion: Sentence starting with “emerging evidence...”, I would expect a couple references at the end of this sentence.

R: This sentence is actually followed by three sentences that provide examples and references (Ref: 10, 14, 15, and 43).

Reviewer: Ok – my original comment is now irrelevant as the sentence appears to have been removed from the discussion.

6. qPCR data: please add all genes analyzed and their expression values as supplementary data to increase transparency of the analysis. As it says in the data availability statement: “All data generated or analyzed in this study are included in this article and the supplementary data file.” I could not find the qPCR data here.

R: We have remade all the bar graphs in Figures 2 – 7. All raw datapoints have been included in the graph.

Reviewer: This specific comment of mine only relates to Figures 6 and 7D, depicting the qPCR results. The plotted black data points are not raw data, instead they appear to be ratios of expression levels of selected target genes between two different treatments. Different absolute gene expression values can lead to the same gene expression ratio when combined, hence why it is so informative to share the underlying gene expression dataset, including both the raw dataset (as it is output by the machine), the normalized dataset (values normalized to e.g. act-1, as the authors describe in the method section), and then finally the calculated fold-changes and expression ratios between treatments that were used

for plotting. A good example of a qPCR dataset shared with best practises in mind can be found here: <https://datadryad.org/stash/dataset/doi:10.5061/dryad.96224rg>. To access the dataset, click on “Download data”, then open the Excel file “qPCR data for Dryad.xlsx”.

7. Typos and phrasing:

o page 3 potassium ion channels

o page 8 bacterial strains

o page 15 “on longer” _ no longer

o page 19 “no long reduce” _ no longer

o page 20 “enter the eukaryotic kingdom” _ enter eukaryotic cells

R: We have fixed all the typos except the last one, which we meant to say that bacterial CDNs enter the eukaryotic kingdom.

Reviewer: Ok – my last point is now anyway irrelevant as the expression appears to have been removed from the manuscript.

8. Needs referencing: page 20, the sentence ending with “ ... and mate.”

R: We have added the reference to this sentence (Ref: 28).

Reviewer: Ok.

Reviewer 3: The revised discussion discusses the findings more broadly, and overall improves the manuscript substantially. The only remaining statement I still have issues with is line 491-493: “This [attraction of worms and infection] would allow *V. cholerae* to colonize the gut of a new host, cause infection, and finally be released into the environment to start the cycle again.”

Since *V. cholerae* in nature is (to the best of my knowledge) exclusively associated with zooplankton, shellfish and other water-dwelling organisms, and the findings do seem to be specific to a soil-dwelling nematode encountering bacteria, I don’t think it is appropriate to speculate about the role of c-di-GMP in the *Vibrio* lifecycle here, since the study does not monitor the response of a natural *Vibrio* host to c-di-GMP. The statement is also not necessary in discussing the main point of the story (c-di-GMP acting as a cue to attract bacterivore nematodes to a food source), so I suggest this statement be removed.

Author’s response: As suggested, we removed this statement.

Reviewer 3: Methods Lifespan: Dead worms were counted and recorded, but also removed?
Author’s response: Yes, the dead worms were removed from the lifespan assay plates after being counted and recorded.

Reviewer 3: Ok. This should be added to the methods description.

Author’s response: We edited the sentence (line 475-476).

Reviewer 3: The grinder section (beginning of page 17: starting with “Younger worms ...”, ending with “... and proliferate”) needs referencing.

Author’s response: This sentence is followed by two sentences, and the references were given at the end of the third one. (Ref: 29, 30)

Reviewer 3: To me as a reader, this reads like refs 29+30 only support the claims on bacterial accumulation and gut immunity/lifespan, and not the claims from the sentences before. Consider adding references after each sentence (just a suggestion).

Author’s response: As suggested, we added the references to this sentence (line 216).

Reviewer 3: This specific comment of mine only relates to Figures 6 and 7D, depicting the qPCR results. The plotted black data points are not raw data, instead they appear to be ratios of expression levels of selected target genes between two different treatments. Different absolute gene expression values can lead to the same gene expression ratio when combined, hence why it is so informative to share the underlying gene expression dataset, including both the raw dataset (as it is output by the machine), the normalized dataset (values normalized to e.g. act-1, as the authors describe in the method section), and then finally the calculated fold-changes and expression ratios between treatments that were used for plotting. A good example of a qPCR dataset shared with best practises in mind can be found here:

<https://datadryad.org/stash/dataset/doi:10.5061/dryad.96224rg>. To access the dataset, click on “Download data”, then open the Excel file “qPCR data for Dryad.xlsx”.

Author’s response: All raw qPCR data were included in the Data source file.